# SSTP: Efficient Sample Selection for Trajectory Prediction

## Abstract

Trajectory prediction is a core task in autonomous driving. However, training advanced trajectory prediction models on existing large-scale datasets is both time-consuming and computationally expensive. More critically, these datasets are highly imbalanced in scenario density, with normal driving scenes (low-moderate traffic) overwhelmingly dominating the datasets, while high-density and safety-critical cases are underrepresented. As a result, models tend to overfit low/moderate-density scenarios and perform poorly in high-density scenarios. To address these challenges, we propose the SSTP framework, which constructs a compact yet density-balanced dataset tailored to trajectory prediction. SSTP consists of two main stages: (1) **Extraction**, where a baseline model is pretrained for a few epochs to obtain stable gradient estimates, and the dataset is partitioned by scenario density. (2) **Selection**, where gradient-based scores and a submodular objective select representative samples within each density category, while biased sampling emphasizes rare high-density interactions to avoid dominance by low-density cases. This approach significantly reduces the dataset size and mitigates scenario imbalance, without sacrificing prediction accuracy. Experiments on the Argoverse 1 and Argoverse 2 datasets with recent state-of-the-art models show that SSTP achieves comparable performance to full-dataset training using only half the data while delivering substantial improvements in high-density traffic scenes and significantly reducing training time. Robust trajectory prediction depends not only on data scale but also on balancing scene density to ensure reliable performance under complex multi agent interactions. The code is available at `https://anonymous.4open.science/r/SSTP_v2-69E5/README.md`.

## 1 Introduction

Trajectory prediction aims to predict the future locations of agents conditioned on their past observations, which plays a key role in the domain of autonomous driving. This task is essential yet challenging due to the complex uncertain driving situations. With rapid developments in deep learning, various methods (Zhou et al., 2022c; Feng et al., 2023; Chai et al., 2019; Gu et al., 2021; Zhang et al., 2021; Zhao et al., 2021; Ngiam et al., 2021) have been proposed with promising trajectory prediction performance. Meanwhile, more large-scale realistic datasets (Caesar et al., 2020; Zhan et al., 2019; Chang et al., 2019; Wilson et al., 2023; Sun et al., 2020) have been released by research institutes and self-driving companies, which further push the boundary of this task.

However, one common issue is that training these data-driven methods requires enormous computational resources and is time-consuming due to the large scale of the datasets. For example, the recent state-of-the-art method MTR (Ngiam et al., 2021) has over 66 million model parameters, and the Waymo Open Motion Dataset (WOMD) (Sun et al., 2020) has over 2.2 million trajectory samples. Training the complete model on this dataset requires a substantial amount of GPU hours, posing a significant computational burden.

*Q1: To what extent does training on massive trajectory datasets improve performance, considering their high computational cost? Q2: How can we effectively reduce the training data volume without significantly compromising model accuracy?*

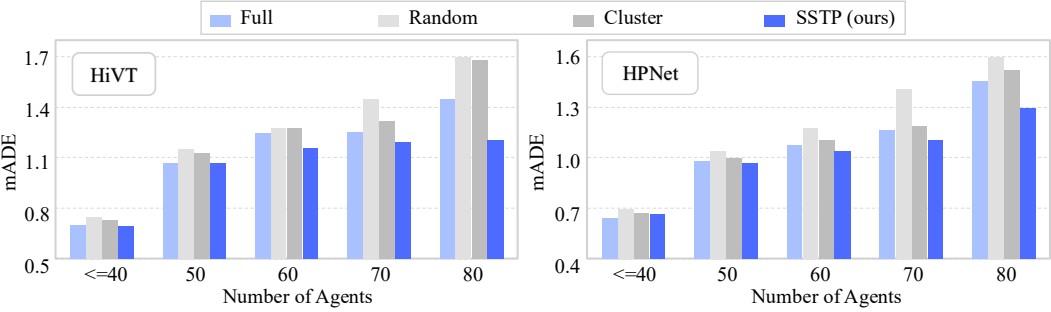

Figure 1: Comparison of minADE across different scenario densities for HiVT and HPNet on Argoverse 1. Models are trained with the full dataset, 50% randomly selected data, 50% cluster-based selected data, and 50% SSTP (ours). SSTP consistently achieves lower errors in high-density scenarios.

Prompted by these questions, we take a deep dive into recent trajectory prediction benchmarks and reveal one finding: the imbalance in the dataset. Specifically, the majority of scenarios involve only a limited number of interacting agents, whereas high-density interaction scenarios are significantly underrepresented. This imbalance is evident in both Argoverse 1 (Chang et al., 2019) and Argoverse 2 (Wilson et al., 2023) (Figure 2), where most cases are low-density. Yet such high-density cases are precisely the most safety-critical for autonomous driving, since they involve complex multi-agent interactions where small prediction errors may directly compromise driving safety. Similar long-tail phenomena have been observed in the literature (Makansi et al., 2021; Wang et al., 2023; Pourkeshavarz et al., 2023; Chen et al., 2024). From a conventional machine learning perspective, standard tasks typically evaluate model performance using the average accuracy across all samples. However, in autonomous driving trajectory prediction, this metric inherently biases models toward driving scenarios with abundant data, leading to suboptimal performance in data-scarce scenarios. For safe driving, an ideal trajectory predictor should exhibit robust performance across both data-rich and data-scarce scenarios. However, due to the imbalanced distribution of driving scenarios, state-of-the-art baselines such as HiVT (Zhou et al., 2022c) and HPNet (Tang et al., 2024) struggle to maintain consistent performance across diverse densities, as illustrated in Figure 1.

To address this issue, we introduce the Sample Selection for Trajectory Prediction (SSTP) framework, the first framework designed to construct a compact and balanced dataset for trajectory prediction. SSTP consists of two main stages: extraction and selection. In the **extraction stage**, a baseline model is pretrained on the full dataset for a few epochs to obtain stable gradient estimates, and the dataset is partitioned by scenario density, measured by the number of agents. This design mitigates the drawbacks of traditional trajectory prediction training: in rare high-density scenarios, the scarcity of training data leads to systematic under-training. For Transformer-based architectures, this issue is particularly pronounced. High-density scenarios require a model to accurately grasp the complex agent–agent dependencies, yet are significantly underrepresented in the training dataset. Consequently, the model does not have sufficient chances to learn such dependencies, leading to poor performance at inference. Moreover, in standard Transformer self-attention, parameters are shared across samples and rely on broad coverage for generalization. When high-density scenarios are scarce, updates are dominated by low-density scenarios, further reinforcing systematic under-training. In the **selection stage**, for each density category, we calculate gradient-based influence scores for every sample, then apply a submodular objective that selects a subset that captures the most representative cases while reducing redundancy. Across partitions, we employ biased sampling to explicitly upweight rare but critical high-density scenarios, preventing dominance by the majority of low-density scenarios. As shown in Figure 1, SSTP reduces data volume while maintaining performance, and achieves even better results in high-density scenarios.

We evaluate our proposed method on the Argoverse 1 (Chang et al., 2019) and Argoverse 2 (Wilson et al., 2023) datasets with multiple baseline methods. Empirical results demonstrate that SSTP successfully constructs a compact and well-balanced dataset. In high-density scenarios, training on the selected subset significantly outperforms using the full dataset, demonstrating the effectiveness of our method. Moreover, our work offers a resource-efficient dataset that maintains balanced performance across various driving scenarios, making it well-suited for training state-of-the-art trajectory prediction models. The main contributions are summarized as follows:

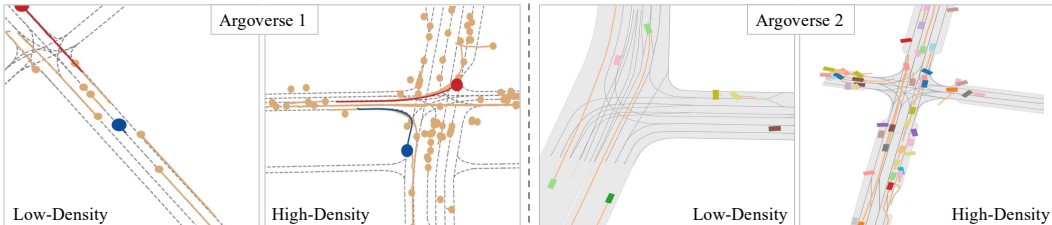

Figure 2: Visualization of different-density scenarios in trajectory prediction datasets. **Left** shows examples from Argoverse 1, and **right** shows examples from Argoverse 2.

- We reveal that modern trajectory prediction datasets are heavily skewed, with scene density and other long-tailed factors biasing models toward majority regimes and limiting performance in safety-critical, rare scenarios.

- We introduce SSTP, a framework that partitions data by scene density and applies submodular selection with gradient-based influence scores, combined with biased sampling to emphasize underrepresented yet safety-critical cases.

- By selecting only 50% of the data, SSTP achieves performance comparable to or better than full-data training, substantially reduces computational cost, reduces error in high-density interaction scenes, and transfers robustly across diverse model architectures.

## 2 METHOD

Denote the training set as $\mathcal{D} = \{S_j\}_{j=1}^M$. Each sample is represented as a triple $S = (X, Y, \mathcal{O})$, where $X$ and $Y$ denote the observed trajectories and future trajectories of all agents, and $\mathcal{O}$ the driving context information (e.g., maps). The goal is to estimate $Y$ conditioned on $X$ and $\mathcal{O}$. We aim to construct a compact subset $\mathcal{C} \subseteq \mathcal{D}$ of target size $B$.

A model trained on $\mathcal{C}$ is expected to achieve trajectory prediction performance comparable to that obtained from training on the full dataset $\mathcal{D}$. Our method is formally described in Algorithm 1. Next, we explain the key algorithmic components in our methods.

**Data Partitioning.** As pointed out in Section 1, data imbalance in agent density is a common characteristic of datasets in the autonomous navigation domain. The impacts of such an imbalance are particularly pronounced when Transformer architectures or GNNs that rely on self-attention or graphs to capture dependencies among agents (Zhou et al., 2022c; 2023; Tang et al., 2024; Zhang et al., 2025). Without density-based partitioning, gradient updates are dominated by abundant low-density scenarios, leading to overfitting on sparse interactions and systematic under-training of high-density scenarios. This imbalance prevents the model from learning robust attention patterns. According to this observation, we first compute a density level $\rho(S_j)$ for each sample $S_j \in \mathcal{D}$ based on the number of agents present. Then, we use a fixed interval $\tau$ to partition the dataset into $K$ disjoint subsets $\mathcal{D}_k$ for $k \in [K]$ based on these density values such that $S \in \mathcal{D}_k$ if $\rho(S) \in [\rho_{\min} + (k-1)\tau, \ \rho_{\min} + k\tau)$, where $\rho_{min}$ is the minimum density level in $\mathcal{D}$.

---

**Algorithm 1** Sample Selection for Trajectory Prediction

**Input:** Full Dataset $\mathcal{D}$, interval $\tau$, ratio $\alpha$, submodular function $P(\cdot)$
**Output:** Target dataset $\mathcal{C}$

1: Initialize: $\mathcal{C} \leftarrow \emptyset$;
2: Initialize: $B \leftarrow \alpha|\mathcal{D}|$;           ▷ Set target size
3: Partitioning: $\mathcal{D}_k$;
4: **For** $k \in \{K, K-1, ..., 1\}$:   ▷ Reverse order
5:    $\mathcal{C}_k \leftarrow \emptyset$;
6:    $n_k \leftarrow$ DynamicSelect$(B, k)$;
7:    **if** $n_k = |\mathcal{D}_k|$ **then**:   ▷ Include all samples
8:        $\mathcal{C} \leftarrow \mathcal{C} \cup \mathcal{D}_k$;
9:    **else**:
10:        **For** $n = 1$ to $n_k$:   ▷ Iterate $n_k$ times
11:            $S_j \leftarrow \arg\min_{S_j \in \mathcal{D}_k \backslash c_k} P(S_j)$;
12:            $\mathcal{C}_k \leftarrow \mathcal{C}_k \cup \{S_j\}$;
13:        **end for**
14:        $\mathcal{C} \leftarrow \mathcal{C} \cup \mathcal{C}_k$;
15:    $B \leftarrow B - |\mathcal{C}_k|$;   ▷ Update remaining size
16: **end for**
17: **Return** $\mathcal{C}$

18: **function** DYNAMICSELECT$(B, k)$:
19:    **if** $|\mathcal{D}_k| \leq \lfloor B/k \rfloor$ **then**:
20:        $n_k \leftarrow |\mathcal{D}_k|$;
21:    **else**:
22:        $n_k \leftarrow \lfloor B/k \rfloor$;
23:    **Return** $n_k$

---

**Gradient Extraction.** We calculate the total loss as follows:

$$\mathcal{L} = \mathcal{L}_{reg} + \mathcal{L}_{cls}, \tag{1}$$

where $\mathcal{L}_{reg}$ measures the L2 norm between the best mode prediction and the ground truth trajectory, and $\mathcal{L}_{cls}$ aligns the mode predicted probabilities $\pi$ with the best mode. By backpropagating loss $\mathcal{L}$, we obtain the gradient with respect to $\hat{\mathbf{Y}}$ as follows:

$$\nabla_{\hat{\mathbf{Y}}} \mathcal{L} = \nabla_{\hat{\mathbf{Y}}} (\mathcal{L}_{reg} + \mathcal{L}_{cls}). \tag{2}$$

Finally, we perform an element-wise multiplication of the calculated gradient with the corresponding decoder latent vector $\mathbf{E}$ as follows:

$$\mathbf{g} = \nabla_{\hat{\mathbf{Y}}} \mathcal{L} \odot \mathbf{E}, \tag{3}$$

where $\mathbf{g}$ captures the joint variations in the gradient and embedding spaces. In this manner, for each subset with $|\mathcal{D}_k| > n_k$, we construct a set of gradient feature vectors $\mathcal{G}_k = \{\mathbf{g}_j\}_{S_j \in \mathcal{D}_k}$ for all trajectory samples in that group.

**Sample Selection.** Recall that $B$ is the target size of the constructed training subset. To ensure fair representation across scene complexities, we adopt a dynamic allocation strategy. For each partition $\mathcal{D}_k$, a local budget $n_k$ is assigned by the `DynamicSelect` function (see Algorithm 1). This mechanism prioritizes high-density subsets that naturally contain fewer samples, preventing them from being underrepresented when the global budget is small. Formally, the allocation satisfies:

$$\sum_{k=1}^{K} n_k = B, \quad n_k \geq 0. \tag{4}$$

Given total $K$ disjoint subsets and the corresponding gradient feature vectors $\mathcal{G}_k$, we present a submodular gain function $P(\cdot)$ to evaluate the contribution of each sample:

$$P(S_j) = \sum_{S_i \in \mathcal{C}_k} \frac{\mathbf{g}_i \cdot \mathbf{g}_j}{\|\mathbf{g}_i\|\|\mathbf{g}_j\|} - \sum_{S_i \in \mathcal{D}_k \setminus \mathcal{C}_k} \frac{\mathbf{g}_i \cdot \mathbf{g}_j}{\|\mathbf{g}_i\|\|\mathbf{g}_j\|}, \tag{5}$$

where we use a cosine similarity kernel to measure the similarity between the sample $S_j$ and other samples. We then apply a greedy optimization strategy, iteratively selecting the sample as follows:

$$S^* = \arg \min_{S_j \in \mathcal{D}_k \setminus \mathcal{C}_k} P(S_j). \tag{6}$$

At each iteration, the selected sample $S^*$ is added to $\mathcal{C}_k$. This process continues until the number of selected samples reaches the budget $n_k$ for subset $\mathcal{D}_k$. Greedy optimization guarantees that the selected set $\mathcal{C}$ satisfies:

$$P(\mathcal{C}) \geq \left(1 - \tfrac{1}{e}\right) P(\mathcal{C}^*), \tag{7}$$

where $\mathcal{C}^*$ denotes the optimal subset under target size $B$. Notably, we process the subsets starting with those having higher density levels, as these subsets tend to be scarce and underrepresented. This prioritization is consistent with our dynamic allocation strategy, which ensures that complex, high-density scenarios are not discarded in early pruning and that long-tailed scenarios remain adequately represented. For subsets $\mathcal{D}_k$ where $|\mathcal{D}_k| \leq n_k$, we directly set $\mathcal{C}_k = \mathcal{D}_k$. Finally, we can yield the target dataset $\mathcal{C} = \bigcup_{k=1}^{K} \mathcal{C}_k$. By incorporating gradient-based similarity into submodular greedy selection, our method maximizes coverage of the gradient space while maintaining diversity, producing a smaller dataset that remains representative and informative. In addition, we account for efficiency: the additional cost of sample selection is dominated by gradient computation and submodular updates. The overall computational complexity can be approximated as $\mathcal{O}(\text{selection}) = \mathcal{O}(|\mathcal{D}| \cdot d) + \mathcal{O}(B \cdot d)$, where $d$ is the dimension of the gradient feature vectors.

**Why Naive Strategies Fall Short.** To motivate the necessity of density-aware, gradient-guided selection, we examine several alternative strategies: ***Re-weighting***: assigns larger loss weights to high-density samples while retaining the full dataset; ***Augmenting***: duplicates high-density samples to artificially increase their proportion; ***High-density+Random*** preserves all high-density samples first and fills the remainder via random sampling; ***Epoch-wise***: dynamically re-selects half the data at the start of each training epoch. These alternatives highlight critical limitations. Both random downsampling and epoch-wise re-selection significantly reduce training data, but the former discards informative scenarios (Table 1, line 4) while the latter introduces instability despite added diversity (line 3). Simple re-weighting or duplication naively equalizes sample presence but often introduces redundancy, showing that effective density balancing requires more than adjusting counts (line 1–2).

Preservation of high-density scenarios is intuitively beneficial, yet without a principled mechanism to regulate redundancy and ensure representativeness, gains remain limited (line 5). Importantly, in trajectory prediction, the scarcity of high-density scenarios is critical, as reliable performance requires learning complex multi-agent interactions that cannot be captured by oversampling or naive mixing. A central observation is that competence in

| Method | #Samples | mADE↓ | mFDE↓ | MR↓ |
|---|---|---|---|---|
| Augmenting | 220k | 0.718 | 1.106 | 0.115 |
| Weighting | 190k | 0.715 | 1.108 | 0.114 |
| Epoch-wise | 95k | 0.752 | 1.189 | 0.130 |
| Random | 95k | 0.741 | 1.164 | 0.125 |
| High-density+Random | 95k | 0.724 | 1.111 | 0.117 |
| **SSTP** | 95k | **0.704** | **1.073** | **0.111** |

Table 1: Comparison with other selection strategies on Argoverse 1. All methods are evaluated at 50% budget (95k samples) unless otherwise specified.

complex, high-density interactions transfers naturally to simpler scenarios as shown in Figure 2, whereas the reverse does not hold. This asymmetry motivates SSTP, which partitions by density and uses gradient-based selection to preserve scarce high-density scenarios while avoiding redundant low-density ones.

## 3 EXPERIMENTS

### 3.1 BENCHMARKS AND SETUP

**Datasets.** We evaluated the effectiveness of our proposed SSTP method on Argoverse Motion Forecasting Dataset 1.1 (Chang et al., 2019) and Argoverse 2 (Wilson et al., 2023). The Argoverse 1 dataset contains 323,557 real-world driving scenarios. All the training and validation scenarios are 5-second sequences sampled at 10 Hz. The length of the historical trajectory for each scenario is 2 seconds, and the length of the predicted future trajectory is 3 seconds. The Argoverse 2 dataset contains 250,000 scenarios, with the same sampling frequency of 10 Hz. Each trajectory has a larger observation window with 5 seconds and a longer prediction horizon with 6 seconds.

**Baselines.** For Argoverse 1, we validate our SSTP method on two SOTA models HiVT (Zhou et al., 2022c) and HPNet (Tang et al., 2024) for evaluation. For Argoverse 2, we evaluate our SSTP method using two SOTA models QCNet (Zhou et al., 2023) and DeMo (Zhang et al., 2025). For a more comprehensive comparison, we also include three following data selection approaches:

(1) Random Selection (Rebuffi et al., 2017): randomly selects a certain proportion of training samples from the original training set.

(2) K-Means Clustering (Likas et al., 2003): clusters trajectories within the observation window based on their features, and then selects the trajectory sample closest to the cluster center in each cluster as a representative.

(3) Herding Selection (Castro et al., 2018): a greedy strategy that first computes the mean feature of all trajectories within the observation window and then iteratively selects trajectory samples that bring the mean of the selected subset as close as possible to the overall mean.

**Metrics.** Following the baselines, we also generate a total 6 future trajectories and use the metrics minimum Average Displacement Error (minADE), minimum Final Displacement Error (minFDE), and Missing Rate (MR) to evaluate the prediction performance.

**Implementation Details.** We primarily utilized the pre-trained HiVT-64 and QCNet as backbone models to perform sample selection on the Argoverse 1 and Argoverse 2 datasets, respectively. To evaluate the performance of the selected subset, we follow their official training and validation protocols. We experimented with different selection ratios, different intervals, and assessed the prediction accuracy of the trajectory models after training on the corresponding subsets. Further implementation details are provided in Appendix A.1.

### 3.2 MAIN RESULTS

We present the primary experimental results in this section, while a comprehensive ablation study, including analyses of alternative strategies and design choices, is provided in Appendix A.3. In addition, qualitative results and visual analyses are presented in Appendix A.4.

| Methods | $\alpha(\%)$ | HiVT-64 | | | HiVT-128 | | | HPNet | | |
|---|---|---|---|---|---|---|---|---|---|---|
| | | mADE↓ | mFDE↓ | MR↓ | mADE↓ | mFDE↓ | MR↓ | mADE↓ | mFDE↓ | MR↓ |
| Argoverse 1 | 100 | **0.695** | **1.037** | **0.109** | **0.666** | **0.978** | **0.091** | **0.647** | **0.871** | **0.070** |
| Random | 60 | 0.745 | 1.163 | 0.132 | 0.719 | 1.078 | 0.129 | 0.680 | 0.951 | 0.091 |
| Cluster | | 0.716 | 1.097 | 0.121 | 0.697 | 1.025 | 0.108 | 0.673 | 0.930 | 0.081 |
| Herding | | 0.723 | 1.101 | 0.125 | 0.685 | 1.018 | 0.106 | 0.666 | 0.922 | 0.085 |
| **SSTP** | | **0.702** | **1.064** | **0.110** | **0.674** | **0.994** | **0.093** | **0.653** | **0.901** | **0.071** |
| Random | 50 | 0.750 | 1.175 | 0.137 | 0.728 | 1.098 | 0.126 | 0.687 | 0.967 | 0.091 |
| Cluster | | 0.725 | 1.117 | 0.124 | 0.692 | 1.033 | 0.118 | 0.676 | 0.952 | 0.085 |
| Herding | | 0.728 | 1.107 | 0.126 | 0.698 | 1.036 | 0.119 | 0.674 | 0.938 | 0.089 |
| **SSTP** | | **0.706** | **1.074** | **0.110** | **0.684** | **1.022** | **0.101** | **0.661** | **0.913** | **0.074** |
| Random | 40 | 0.752 | 1.183 | 0.139 | 0.727 | 1.109 | 0.126 | 0.696 | 0.987 | 0.099 |
| Cluster | | 0.732 | 1.141 | 0.127 | 0.703 | 1.058 | 0.121 | 0.681 | 0.962 | 0.089 |
| Herding | | 0.722 | 1.123 | 0.128 | 0.704 | 1.056 | 0.119 | 0.684 | 0.956 | 0.093 |
| **SSTP** | | **0.711** | **1.088** | **0.114** | **0.696** | **1.048** | **0.106** | **0.671** | **0.931** | **0.076** |

Table 2: Performance comparison results on Argoverse 1 with data retention ratios of 60%, 50%, and 40%. The compared methods include Random Selection, K-Means Clustering, and Herding Selection. The model used for data selection is HiVT-64, while the evaluation is conducted on HiVT-64, HiVT-128, and HPNet. $\alpha$ (%) represents the proportion of retained data relative to the full training set.

| | Agent<40 | | | Agent>=40 | | | Agent>=60 | | | Agent>=80 | | |
|---|---|---|---|---|---|---|---|---|---|---|---|---|
| | mADE↓ | mFDE↓ | MR↓ | mADE↓ | mFDE↓ | MR↓ | mADE↓ | mFDE↓ | MR↓ | mADE↓ | mFDE↓ | MR↓ |
| Full | **0.700** | **1.071** | **0.108** | **0.950** | **1.456** | **0.171** | 1.248 | 1.898 | 0.283 | 1.450 | 2.059 | 0.361 |
| Random | 0.734 | 1.127 | 0.119 | 0.997 | 1.552 | 0.193 | 1.287 | 1.989 | 0.315 | 1.638 | 2.359 | 0.389 |
| Cluster | 0.719 | 1.121 | 0.115 | 0.971 | 1.501 | 0.188 | 1.251 | 1.898 | 0.304 | 1.568 | 2.257 | 0.376 |
| Herding | 0.726 | 1.122 | 0.116 | 0.980 | 1.528 | 0.190 | 1.258 | 1.922 | 0.310 | 1.581 | 2.381 | 0.372 |
| **SSTP** | 0.714 | 1.098 | 0.113 | 0.962 | 1.497 | 0.183 | **1.219** | **1.835** | **0.280** | **1.373** | **1.762** | **0.277** |

Table 3: Comparison of model performance across scenario densities when trained on the full dataset, a 50% random subset, and a 50% SSTP subset, where SSTP consistently achieves superior results.

Table 2 showcases the strong performance of our selected subset on the Argoverse 1 dataset across all compression rates. Following the same experimental setup as the baseline models, we trained HiVT and HPNet from scratch on the subset. Our method significantly reduces data volume while maintaining nearly lossless performance. Even with only half of the data, models trained on our selected subset still perform comparably to those trained on the full dataset and consistently outperform random selection, clustering, and herding. Furthermore, our subset also demonstrates strong results on larger models such as HiVT-128 and HPNet. Additional results for other data retention ratios are provided in Appendix A.2.1, with detailed numbers reported in Table 6.

**Performance Enhancement.** Scene density in autonomous driving varies substantially, yet most existing trajectory prediction datasets are dominated by low-density scenarios. From a safety perspective, however, an ideal trajectory predictor must perform reliably across the full spectrum of scene complexities. To this end, we evaluate our proposed method on multiple models and across different density levels. As shown in Table 3, our method consistently outperforms models trained on the full dataset, particularly in high-density scenarios, on both the Argoverse 1 and Argoverse 2 datasets. In low-density settings (fewer than 40 agents), our selected subset achieves performance nearly identical to the full dataset, with only marginal increases of minADE and minFDE, while MR remains almost unchanged, which is negligible given the limited interactions in such scenes.

When the agent density increases, our method brings the most substantial gains. Compared to all other baselines, SSTP achieves lower displacement errors and notably reduces MR. For example, when the number of agents exceeds 80, SSTP cuts the missing rate by more than 8% relative to random selection, and also outperforms clustering and herding by clear margins. These results highlight that while clustering and herding provide partial improvements by ensuring representativeness, they are still insufficient for handling highly complex traffic scenes. In contrast, SSTP effectively

| Method | $\alpha$(%) | QCNet | | | DeMo | | |
|---|---|---|---|---|---|---|---|
| | | mADE↓ | mFDE↓ | MR↓ | mADE↓ | mFDE↓ | MR↓ |
| Argoverse 2 | 100 | **0.724** | **1.258** | **0.162** | **0.657** | **1.254** | **0.163** |
| Random | | 0.787 | 1.419 | 0.208 | 0.755 | 1.433 | 0.198 |
| Cluster | 60 | 0.773 | 1.406 | 0.192 | 0.693 | 1.386 | 0.187 |
| Herding | | 0.778 | 1.402 | 0.191 | 0.701 | 1.494 | 0.189 |
| **SSTP** | | **0.740** | **1.316** | **0.163** | **0.682** | **1.344** | **0.164** |
| Random | | 0.805 | 1.447 | 0.219 | 0.756 | 1.448 | 0.203 |
| Cluster | 50 | 0.798 | 1.435 | 0.193 | 0.732 | 1.437 | 0.191 |
| Herding | | 0.782 | 1.407 | 0.190 | 0.731 | 1.434 | 0.190 |
| **SSTP** | | **0.754** | **1.352** | **0.172** | **0.704** | **1.414** | **0.173** |
| Random | | 0.811 | 1.471 | 0.226 | 0.763 | 1.475 | 0.202 |
| Cluster | 40 | 0.813 | 1.495 | 0.214 | 0.732 | 1.456 | 0.195 |
| Herding | | 0.807 | 1.454 | 0.198 | 0.739 | 1.460 | 0.197 |
| **SSTP** | | **0.778** | **1.410** | **0.183** | **0.723** | **1.450** | **0.191** |

Table 4: Performance comparison results on Argoverse 2 with data retention ratios of 60%, 50%, and 40%. The compared methods include Random Selection and K-means clustering. The model used for data selection is QCNet, while the evaluation is conducted on QCNet and DeMo.

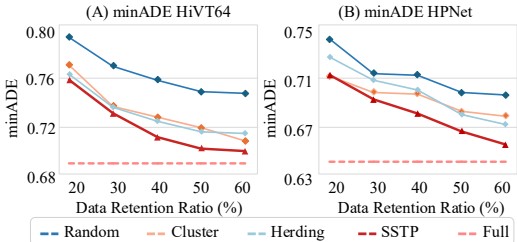 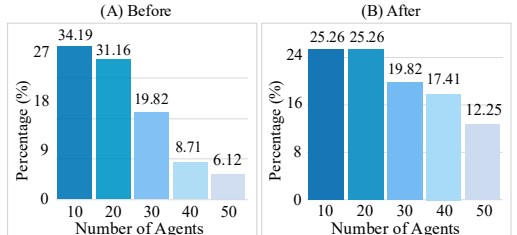

Figure 3: Performance of HiVT and HPNet trained on subsets selected by different methods at varying retention ratios, evaluated with minADE.

Figure 4: Distribution of scenarios categorized by agent density in Argoverse 1 before (**left**) and after (**right**) applying SSTP with 50% data retention ratio.

balances scene density while selecting informative samples, leading to consistently superior performance across all density levels. Similarly, the SOTA model HPNet, when trained on our selected subset, even outperforms its counterpart trained on the full dataset. Detailed results are provided in Appendix A.2.2, with comprehensive numbers reported in Table 8.

**Generalization Across Datasets.** We further evaluated our proposed method on the Argoverse 2 dataset, which presents greater challenges due to its more diverse driving scenarios and longer prediction horizons. As shown in Table 4, our method consistently outperforms other data selection strategies across all data retention rates, achieving lower minADE and minFDE while maintaining a lower MR. These results further validate the robustness of our approach, as it maintains strong performance across different datasets. This demonstrates that our method is not only effective within a specific dataset but also generalizes well to more complex and diverse trajectory scenarios, such as those found in Argoverse 2. Additional results for other data retention ratios are provided in Appendix A.2.1, with detailed numbers reported in Table 7.

**Data Retention Ratio.** To examine the impact of different data retention ratios on model performance, we conducted experiments with retention rates of $\alpha = 60, 50, 40, 30, 20, 10\%$ as shown in Figure 3. When higher model performance is required, retaining 50% of the data already achieves results comparable to training with the full dataset. This demonstrates the effectiveness of our SSTP method, as the selected subset is of higher quality compared to equally sized subsets chosen by other methods. Furthermore, under limited computational resources, even retaining only 20% of the data still yields reasonably good results. We further analyze the behavior under extremely low retention ratios, with detailed results provided in Appendix A.2.3 and Table 9.

| Variants | Data Distribution(%) | | Model Performance | | |
|---|---|---|---|---|---|
| | Agent<40 | Agent>=40 | mADE↓ | mFDE↓ | MR↓ |
| Full dataset | 93.88 | 6.12 | 0.692 | 1.047 | 0.104 |
| Random | 85.16 | 14.84 | 0.741 | 1.164 | 0.125 |
| SSTP w/ Submodular | 93.88 | 6.12 | 0.724 | 1.115 | 0.116 |
| SSTP w/ Partition | 70.35 | 29.65 | 0.729 | 1.116 | 0.118 |
| **SSTP** | 70.35 | 29.65 | **0.704** | **1.073** | **0.111** |

Table 5: Performance comparison of data selection strategies on HiVT trained with Argoverse 1. This table shows the impact of partitioning and selection with Submodular Gain strategies on data distribution and model performance. The whole dataset and random selection serve as baselines, while different variations of SSTP are evaluated. Our method (SSTP), integrates both strategies, achieves the best results by maintaining a balanced data distribution and reducing minADE, minFDE, and MR.

**Density Balancing.** Our method explicitly controls scene density distribution during selection, ensuring a more balanced dataset, as illustrated in Figure 4. In contrast, random selection fails to maintain this balance, leading to uneven representation of scenarios with varying complexity and weaker generalization. As shown in Table 5 line 4, applying scene balancing alone already improves performance compared to random selection, demonstrating that controlling scene density enhances the effectiveness of trajectory prediction models. However, balancing alone remains insufficient. By further integrating submodular selection to account for sample informativeness, our method achieves the best performance across all metrics. These findings indicate that while scene balancing is beneficial, it is insufficient to achieve optimal performance without also considering sample informativeness.

**Efficiency.** Our method significantly reduces computational time while maintaining strong performance, as shown in Figure 5. Training the HiVT model on the full dataset requires 7.78 hours. In contrast, utilizing our SSTP method to select a 60% subset requires only 0.6 hour for pre-training and 2.30 hours for selection, reducing the overall training time. When training on the selected subset, the total training time

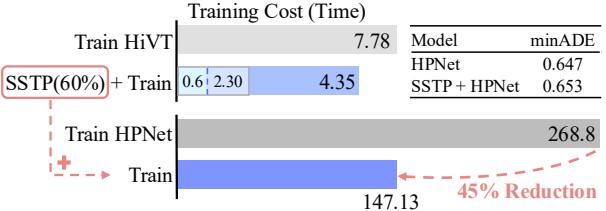

Figure 5: Training time comparison of HiVT and HPNet. Our method achieves over 50% reduction in time while maintaining comparable accuracy.

decreases to 7.25 hours (0.6 + 2.30 + 4.35), with only a minor increase of 0.007 in minADE.

Beyond efficiency on the original backbone, a key advantage of SSTP is that the selected subset is model-agnostic and can be directly reused to train other trajectory predictors. For instance, when applied to HPNet, training on the full dataset takes 307.2 hours, whereas training with the selected subset reduces the time by over 100 hours, cutting the training cost by nearly 45%, while achieving comparable or even better accuracy. This decisively confirms the superior efficiency of our approach in balancing training cost and model performance.

**Generalizability across Backbones.** To further assess the generalizability of our method, we evaluated its performance with different backbone models. Specifically, we used pre-trained HiVT-64 and HPNet as feature extractors on the Argoverse 1 dataset and conducted experiments under varying data retention ratios. Detailed results are provided in Appendix A.2.4, with numerical values reported in Table 10. As shown, regardless of the backbone used for subset selection, the final trajectory prediction performance remains nearly identical. This demonstrates that our subset selection strategy is largely backbone-agnostic, underscoring its robustness and broad applicability for optimizing diverse trajectory prediction models.

# 4 RELATED WORK

**Trajectory Prediction.** Trajectory prediction infers an agent's future motion from its historical observations. In recent years, research has increasingly concentrated on capturing complex multi-

agent interactions, driving advances in various predictive methods (Wang et al., 2024; Giuliari et al., 2021; Xu et al., 2020; Liu et al., 2021; Ngiam et al., 2021; Zhou et al., 2022c; 2023; Tang et al., 2024; Zhang et al., 2025; Liang et al., 2020). Furthermore, novel approaches including pretraining (Chen et al., 2023; Cheng et al., 2023; Lan et al., 2023), historical prediction structure design (Park et al., 2024; Tang et al., 2024), GPT-style next-token prediction (Philion et al., 2023; Seff et al., 2023), and post-processing optimization (Zhou et al., 2024; Choi et al., 2023) have significantly enhanced model performance, demonstrating strong results across various datasets. However, most state-of-the-art methods rely on training with large-scale datasets (Caesar et al., 2020; Chang et al., 2019; Sun et al., 2020; Wilson et al., 2023; Ettinger et al., 2021), leading to significant computational costs. In contrast, we propose a pioneering data selection strategy in the trajectory prediction domain that constructs a compact, balanced, yet highly representative dataset, significantly reducing training time while preserving model performance.

**Long-Tail in Trajectory Prediction Dataset** The performance of the trajectory prediction models is evaluated on the overall average. While they excel on benchmarks, these models often struggle with challenging scenarios (Makansi et al., 2021; Wang et al., 2023; Pourkeshavarz et al., 2023) due to the long-tail data distribution, where common cases dominate and complex or rare situations are underrepresented (Chen et al., 2024). Recent studies (Wang et al., 2023; Zhang et al., 2024; Zhou et al., 2022a; Lan et al., 2024) have started addressing the long-tail problem in trajectory prediction, primarily by leveraging contrastive learning to enhance feature representations. However, these methods mainly focus on optimizing feature-level learning while overlooking scenario level distribution and the importance of individual samples. In contrast, our proposed method assesses the contribution of each sample and applies a refined selection strategy to build a balanced and compact dataset. Experimental results show that it significantly boosts performance in complex scenarios.

**Training Sample Selection** Deep neural networks, especially Transformers, depend on large datasets and incur high computational costs. To reduce these costs and shorten training time, various methods to improve data efficiency have been proposed, including frequent parameter updates (Robbins & Monro, 1951), fewer iterations (Sutskever et al., 2013), and dynamic learning rate adjustments (Kingma, 2014; Duchi et al., 2011). To directly reduce data volume, dataset condensation compresses raw data into compact synthetic samples (Wang et al., 2018; Zhao et al., 2020; Zhao & Bilen, 2021; Kim et al., 2022; Wang et al., 2022; Zhao & Bilen, 2023; Cazenavette et al., 2022). Another widely studied approach is coreset selection, which constructs a weighted subset that closely approximates the statistical distribution of the original dataset (Har-Peled & Mazumdar, 2004; Coleman et al., 2019; Margatina et al., 2021; Mirzasoleiman et al., 2020). However, those approaches on dataset condensation (Nguyen et al., 2020; Loo et al., 2022; Zhou et al., 2022b; Sajedi et al., 2023; Zhao et al., 2023) and coreset selection (Killamsetty et al., 2021; Paul et al., 2021) has been predominantly focused on image classification. In contrast, we propose a sample selection strategy based on submodular functions for the trajectory prediction domain. As a pioneering method, our approach significantly reduces the training data required while maintaining model performance.

## 5 CONCLUSION

In this paper, we presented the Sample Selection for Trajectory Prediction (SSTP) framework, a novel, data-centric approach that constructs a compact yet balanced dataset for trajectory prediction. SSTP effectively tackles the challenges posed by data imbalance and the high training costs inherent in large-scale trajectory datasets. By significantly reducing the training data volume while maintaining, and even enhancing the model performance in high-density scenarios. SSTP not only accelerates training but also delivers results comparable to, or better than, those achieved with full-dataset training. Extensive evaluations on the Argoverse 1 and Argoverse 2 benchmarks across a wide range of state-of-the-art models underscore the practical value of our approach in improving both efficiency and robustness in trajectory prediction for autonomous driving.

**Limitations.** While effective, our method still incurs non-negligible computational overhead during the gradient extraction and submodular optimization stages of sample selection. Although this is a one-time cost, further optimization to reduce its complexity would be desirable. Moreover, model performance degrades under extremely low data retention ratios (e.g., 10%), highlighting the challenge of preserving robustness when only very limited data are available. Finally, the present study is restricted to trajectory prediction tasks on the Argoverse benchmarks; extending SSTP to other autonomous driving tasks and additional datasets remains an important avenue for future research.

ETHICS STATEMENT

This work uses only publicly available autonomous driving datasets (Argoverse 1 and 2) containing vehicle trajectories and maps, with no human subjects or private data involved. We adhere to the conference Code of Ethics and confirm full compliance with its standards.

REPRODUCIBILITY STATEMENT

Implementation details, including dataset preprocessing, training protocols, and hyperparameter settings, are provided in Section 4 and Appendix A.1. We also release our code and checkpoints (via the anonymous link) to facilitate full reproducibility.

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

# A APPENDIX

## A.1 IMPLEMENTATION DETAILS

During the dataset selection stage, we employ HiVT (Zhou et al., 2022c), HPNet (Tang et al., 2024), and QCNet (Zhou et al., 2023) as backbones with batch sizes of 16, 4, and 4, respectively, selected using the NaiveGreedy algorithm. For model evaluation, one mainstream model and one SOTA model are chosen per dataset: HiVT and HPNet for Argoverse 1 (Chang et al., 2019), and QCNet and DeMo (Zhang et al., 2025) for Argoverse 2 (Wilson et al., 2023). All models follow their official experimental settings and are trained on both selected subsets and complete datasets.

**HiVT**: Batch size 32, LR $3 \times 10^{-4}$, weight decay $1 \times 10^{-4}$, dropout 0.1. Its architecture includes 1 interaction module, 4 temporal and 3 global modules, 8-head attention (50 m radius), 6 prediction modes, and hidden sizes of 64/128.

**HPNet**: Batch size 16, LR $5 \times 10^{-4}$ (same weight decay and dropout). It has 1 spatiotemporal and 2 tri-factor attention layers (50 m radius, 20-frame windows) with data augmentation: horizontal flipping (0.5), agent (0.05) and lane (0.2) occlusion.

**QCNet**: Trained for 64 epochs with AdamW (batch size 32, LR $5 \times 10^{-4}$, weight decay $1 \times 10^{-4}$, dropout 0.1). It uses a 128-dim hidden space, gated 8-head attention, layer normalization on MLP and attention layers, 3 iterations for the trajectory proposal, and 2 multi-context attention modules in both encoder and decoder.

**DeMo**: Trained for 60 epochs with AdamW (batch size 16 per GPU, LR 0.003, weight decay 0.01, cosine annealing with 10 warm-up epochs). No data augmentation is applied.

## A.2 ADDITIONAL EXPERIMENTS

### A.2.1 ADDITIONAL RESULTS ON ARGOVERSE 1 AND ARGOVERSE 2.

We present comprehensive results for different models and data retention ratios on SSTP-selected datasets for both Argoverse 1 and Argoverse 2. The results, shown in Table 6 and Table 7, follow trends consistent with those observed in the 60%, 50%, and 40% retention settings, further validating the effectiveness of our approach.

| Methods | $\alpha$(%) | HiVT-64 | | | HiVT-128 | | | HPNet | | |
|---|---|---|---|---|---|---|---|---|---|---|
| | | mADE↓ | mFDE↓ | MR↓ | mADE↓ | mFDE↓ | MR↓ | mADE↓ | mFDE↓ | MR↓ |
| Random | 30 | 0.76 | 1.21 | 0.13 | 0.73 | 1.13 | 0.12 | 0.69 | 0.98 | 0.09 |
| Cluster | | 0.74 | 1.14 | 0.12 | 0.71 | 1.07 | 0.11 | 0.68 | 0.97 | 0.08 |
| Herding | | 0.73 | 1.14 | 0.12 | 0.71 | 1.07 | 0.11 | 0.69 | 0.96 | 0.09 |
| **SSTP** | | **0.72** | **1.12** | **0.12** | **0.70** | **1.06** | **0.11** | **0.68** | **0.95** | **0.07** |
| Random | 20 | 0.78 | 1.25 | 0.14 | 0.75 | 1.18 | 0.13 | 0.71 | 1.01 | 0.10 |
| Cluster | | 0.76 | 1.21 | 0.14 | 0.72 | 1.11 | 0.12 | 0.69 | 0.98 | 0.09 |
| Herding | | 0.76 | 1.19 | 0.13 | 0.74 | 1.14 | 0.13 | 0.70 | 0.99 | 0.09 |
| **SSTP** | | **0.74** | **1.16** | **0.12** | **0.72** | **1.11** | **0.11** | **0.69** | **0.97** | **0.08** |
| Random | 10 | 0.84 | 1.40 | 0.16 | 0.80 | 1.29 | 0.15 | 0.88 | 1.40 | 0.18 |
| Cluster | | 0.82 | 1.37 | 0.16 | 0.76 | 1.20 | 0.13 | 0.87 | 1.39 | 0.17 |
| Herding | | 0.80 | 1.29 | 0.14 | 0.80 | 1.29 | 0.14 | 0.81 | 1.26 | 0.13 |
| **SSTP** | | **0.78** | **1.25** | **0.13** | **0.75** | **1.17** | **0.12** | **0.78** | **1.17** | **0.12** |

Table 6: Performance comparison results on Argoverse 1 with data retention ratios of 30%, 20%, and 10%. Pretrained HiVT-64 is used for sample selection. Evaluation conducted on HiVT-64, HiVT-128, and HPNet.

### A.2.2 PERFORMANCE ENHANCEMENT ON HPNET.

To further validate the generalization capability of the SSTP method, we conducted the same experimental setup on HPNet. The test set was partitioned based on different scene densities, and the model was evaluated on these subdivided datasets. The results, presented in Table 8, demonstrate

| Methods | $\alpha$(%) | QCNet | | | DeMo | | |
|---|---|---|---|---|---|---|---|
| | | mADE↓ | mFDE↓ | MR↓ | mADE↓ | mFDE↓ | MR↓ |
| Random | | 0.827 | 1.486 | 0.224 | 0.779 | 1.547 | 0.216 |
| Cluster | 30 | 0.821 | 1.523 | 0.222 | 0.765 | **1.498** | 0.207 |
| Herding | | 0.819 | 1.505 | 0.213 | 0.762 | 1.506 | 0.204 |
| **SSTP** | | **0.794** | **1.453** | **0.192** | **0.743** | 1.501 | **0.192** |
| Random | | 0.878 | 1.586 | 0.242 | 0.827 | 1.645 | 0.238 |
| Cluster | 20 | 0.843 | 1.561 | 0.222 | 0.794 | 1.618 | 0.229 |
| Herding | | 0.844 | 1.545 | 0.222 | 0.801 | 1.625 | 0.227 |
| **SSTP** | | **0.832** | **1.530** | **0.211** | **0.783** | **1.591** | **0.212** |
| Random | | 0.944 | 1.797 | 0.272 | 0.918 | 1.817 | 0.263 |
| Cluster | 10 | 0.911 | 1.772 | 0.251 | 0.878 | 1.779 | 0.247 |
| Herding | | 0.908 | 1.743 | 0.246 | 0.882 | 1.774 | 0.243 |
| **SSTP** | | **0.891** | **1.684** | **0.241** | **0.871** | **1.770** | **0.241** |

Table 7: Performance comparison results on Argoverse 2 with data retention ratios of 30%, 20%, and 10%. Pretrained QCNet is used for sample selection. Evaluation conducted on QCNet and DeMo.

| HPNet | Agent<40 | | | Agent>=40 | | | Agent>=60 | | | Agent>=80 | | |
|---|---|---|---|---|---|---|---|---|---|---|---|---|
| | mADE↓ | mFDE↓ | MR↓ | mADE↓ | mFDE↓ | MR↓ | mADE↓ | mFDE↓ | MR↓ | mADE↓ | mFDE↓ | MR↓ |
| Full | 0.611 | 0.833 | 0.062 | 0.875 | 1.202 | 0.127 | 1.111 | 1.463 | 0.201 | 1.596 | 1.771 | 0.276 |
| Random | 0.652 | 0.903 | 0.071 | 0.932 | 1.336 | 0.147 | 1.267 | 1.850 | 0.274 | 1.649 | 2.086 | 0.375 |
| Cluster | 0.642 | 0.889 | 0.069 | 0.916 | 1.291 | 0.143 | 1.193 | 1.669 | 0.250 | 1.636 | 1.977 | 0.325 |
| Herding | 0.650 | 0.901 | 0.070 | 0.921 | 1.303 | 0.145 | 1.215 | 1.673 | 0.252 | 1.637 | 2.001 | 0.342 |
| **SSTP** | **0.630** | **0.861** | **0.064** | 0.878 | 1.210 | 0.132 | **1.110** | **1.461** | **0.198** | **1.574** | **1.698** | **0.241** |

Table 8: Performance Comparison of HPNet across different scene densities when trained on different sample set with $\alpha = 50\%$.

the effectiveness of our approach. When the agent density is below 40, our method achieves performance comparable to models trained on the full dataset, with only slight increases of 0.019 in minADE and 0.028 in minFDE, while MR remains nearly unchanged. However, in high-density scenarios where the number of agents exceeds 60, models trained on our selected subset exhibit notable improvements, with minADE and minFDE reduced by 0.001 and 0.002, respectively. This advantage becomes even more pronounced in scenarios with more than 80 agents, where minADE is reduced by approximately 0.022, minFDE by nearly 0.073, and MR by nearly 3.5%. These findings confirm that SSTP effectively maintains dataset diversity while preserving model performance, regardless of the model used.

### A.2.3    LOW DATA RETENTION SETTING.

We further studied the extremely low-budget regime with $\alpha \in \{10, 5, 2, 1\}\%$. As shown in Table 9, the performance of all methods drops sharply below 10%. Nevertheless, our SSTP method consistently outperforms random selection at the same budget levels. In particular, at 1% retention, SSTP reduces minADE from 2.033 to 1.589 and MR from 0.452 to 0.386, showing that even with very limited data, informative subset selection remains beneficial. This advantage comes from prioritizing scarce high-density scenes during budget allocation, ensuring long-tailed scenarios remain represented.

### A.2.4    GENERALIZABILITY AND ROBUSTNESS.

To further assess the generalizability and robustness of our proposed method, we evaluate the impact of different data retention ratios on SSTP-selected datasets using various backbone networks. The results, presented in Table 10, demonstrate that SSTP is not only adaptable across a diverse range of trajectory prediction models but also effectively reduces dataset size while maintaining model performance across different feature extractors. These findings underscore SSTP's robustness and broad applicability in real-world autonomous driving scenarios.

| $\alpha(\%)$ | Method | mADE↓ | mFDE↓ | MR↓ |
|---|---|---|---|---|
| 10 | Random | 0.818 | 1.329 | 0.167 |
| | **SSTP** | **0.781** | **1.253** | **0.132** |
| 5 | Random | 0.869 | 1.484 | 0.183 |
| | **SSTP** | **0.843** | **1.416** | **0.161** |
| 2 | Random | 1.055 | 1.902 | 0.245 |
| | **SSTP** | **1.015** | **1.871** | **0.220** |
| 1 | Random | 2.033 | 4.613 | 0.452 |
| | **SSTP** | **1.589** | **3.416** | **0.386** |

Table 9: Performance comparison under extremely low data retention settings ($\alpha \leq 10\%$). For each retention ratio, SSTP consistently outperforms random selection, though performance degrades sharply when the data retention ratio drops below 10%.

| Backbone | $\alpha(\%)$ | HiVT-64 | | | HPNet | | |
|---|---|---|---|---|---|---|---|
| | | mADE↓ | mFDE↓ | MR↓ | mADE↓ | mFDE↓ | MR↓ |
| HiVT-64 | 10 | 0.786 | 1.254 | 0.138 | 0.782 | 1.172 | 0.121 |
| | 20 | 0.749 | 1.167 | 0.125 | 0.691 | 0.970 | 0.081 |
| | 30 | 0.727 | 1.120 | 0.120 | 0.678 | 0.951 | 0.079 |
| | 40 | 0.712 | 1.089 | 0.114 | 0.671 | 0.931 | 0.076 |
| | 50 | 0.704 | 1.073 | 0.111 | 0.661 | 0.913 | 0.074 |
| | 60 | 0.703 | 1.065 | 0.111 | 0.654 | 0.901 | 0.072 |
| HPNet | 10 | 0.796 | 1.277 | 0.143 | 0.739 | 1.050 | 0.095 |
| | 20 | 0.753 | 1.193 | 0.125 | 0.699 | 0.975 | 0.082 |
| | 30 | 0.732 | 1.143 | 0.119 | 0.681 | 0.943 | 0.078 |
| | 40 | 0.717 | 1.087 | 0.112 | 0.670 | 0.924 | 0.075 |
| | 50 | 0.708 | 1.079 | 0.111 | 0.664 | 0.917 | 0.074 |
| | 60 | 0.703 | 1.067 | 0.108 | 0.657 | 0.910 | 0.075 |

Table 10: Performance comparison of different backbone models, HiVT-64 and HPNet, trained on subsets selected by different strategies at varying data retention ratios.

## A.3 ABLATION STUDY

**Effectiveness of Submodular Gain.** To isolate the effect of submodular gain, we conducted an experiment where data selection was based solely on submodular importance scores, without considering scene density balancing, as shown in Table 5 line 3. The results indicate that using only submodular selection achieves a minADE of 0.724, lower than the 0.741 obtained through random selection, demonstrating that submodular-based sample selection improves data quality. However, it still underperforms compared to our full method. This is because prioritizing sample informativeness without adjusting for scene density leads to a dataset biased toward certain complexity levels, ultimately hindering the model's generalization ability. In contrast, our full method, which integrates scene balancing with submodular gain, achieves the best performance across all metrics. These findings highlight the necessity of jointly considering both scene distribution balance and sample informativeness to construct an effective training dataset.

**Impact of Pretrained Backbone Epochs.** To examine the influence of the pretrained backbone on subset selection, we conducted a series of experiments using models initialized identically but trained with different numbers of pretraining epochs. Taking HiVT-64 as an example, the official training setup involves training the model for 64 epochs using the full dataset. In our experiments, we varied the number of pretraining epochs as $\{0, 5, 8, 10, 15, 64\}$ and analyzed its impact on subset selection, as shown in Table 11. The results indicate that moderate pretraining is crucial for effective subset selection. When the number of pretraining epochs is set to 5, the subset selection achieves optimal performance, consistently outperforming other configurations across all data retention ratios. As the pretraining epochs increase, subset selection continues to provide significant advantages over other data selection methods but does not surpass the performance observed at epoch 5. For models without pretraining, minADE degrades noticeably compared to models pretrained for 5 epochs. In

| Epoch | 0 | 5 | 8 | 10 | 15 | 64 |
|---|---|---|---|---|---|---|
| mADE↓ | 0.713 | **0.704** | 0.707 | 0.708 | 0.710 | 0.712 |
| mFDE↓ | 1.083 | **1.073** | 1.076 | 1.074 | 1.083 | 1.080 |
| MR↓ | 0.112 | **0.111** | 0.111 | 0.111 | 0.111 | 0.111 |

Table 11: Performance comparison of models pretrained on the full dataset for different numbers of epochs. The pretrained weights are then used to initialize the model for SSTP, selecting 50% of the data.

| Partition | mADE↓ | mFDE↓ | MR↓ |
|---|---|---|---|
| $\tau = 5$ | 0.703 | **1.056** | **0.110** |
| $\tau = 10$ | **0.702** | 1.064 | 0.111 |
| $\tau = 20$ | 0.707 | 1.081 | 0.113 |

Table 12: Comparison of SSTP with different partition intervals $\tau$ and its impact on HiVT performance using the Argoverse 1.

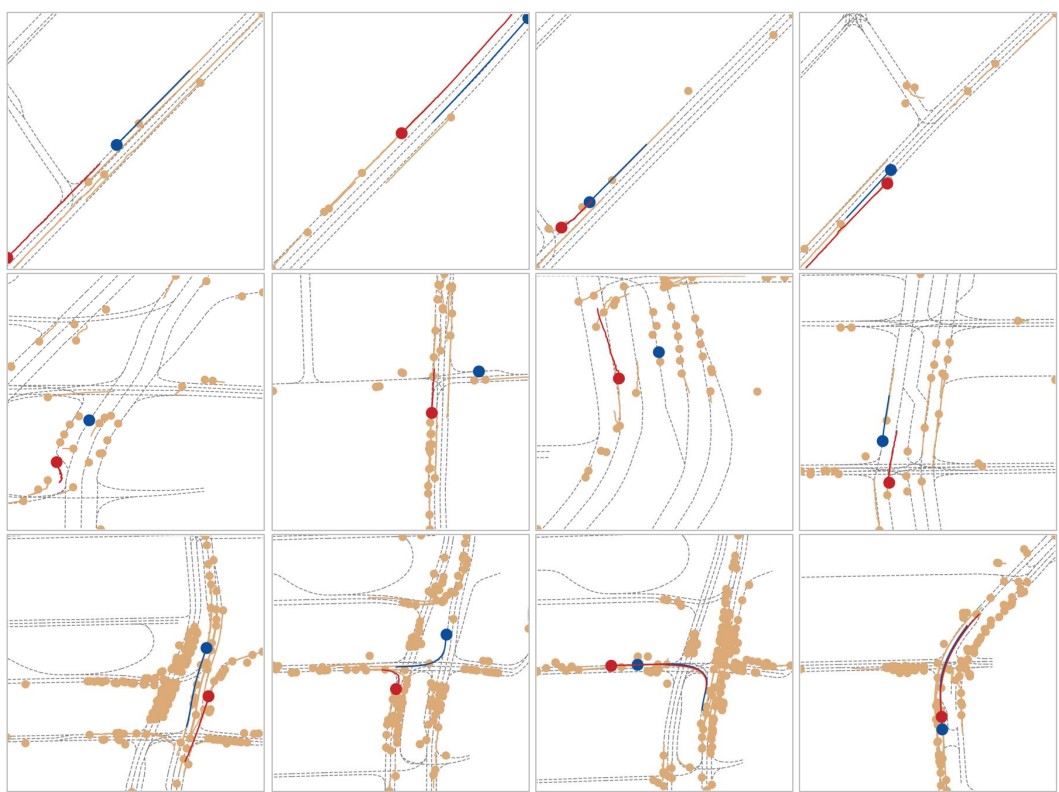

Figure 6: Visualization of different agent density scenarios in Argoverse 1.

contrast, at 64 pretraining epochs, as the model has already converged, the impact of sample selection on gradient updates diminishes. Although subset selection performance remains competitive, it does not yield further improvements over moderate pretraining.

**Partition Interval.** Given the number of agents in different trajectory prediction scenarios varies significantly, we examine the impact of scene density partition intervals on the selected subset to validate the generalizability of our method. Specifically, we divide the scenes in the Argoverse 1 dataset based on agent counts with partition intervals $\tau$ of {5, 10, 20}, forming multiple scene density categories. Within each category, we perform data selection. As shown in Table 12, the subsets selected using different partition intervals result in comparable model performance, with minimal variations in minADE and minFDE. This consistency across different settings highlights the robustness of our method. This suggests that our approach generalizes well to other datasets. When applying this method, the partition interval can be adjusted based on the dataset characteristics: if the dataset has relatively few agents per scene, a smaller interval is preferable; whereas for datasets with a high variance in agent count, a larger interval may be more suitable to better balance scene density.

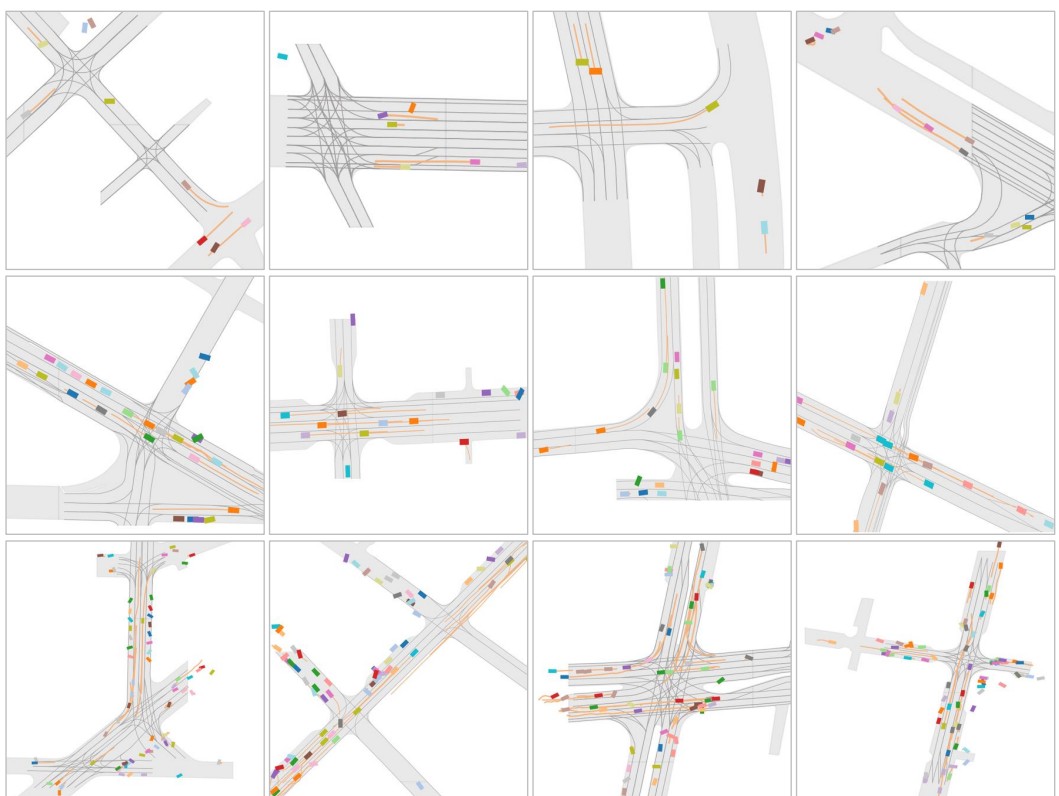

Figure 7: Visualization of different agent density scenarios in Argoverse 2.

## A.4 QUALITATIVE RESULTS

In Figure 6 and Figure 7, we present trajectory samples from three different scene densities in Argoverse 1 and Argoverse 2. In the top row, the scenes feature only a few agents, making them relatively straightforward for the model to predict. Because low-density scenarios dominate the dataset, the model tends to be biased toward these simpler cases, and the lower interaction complexity naturally reduces motion-forecasting uncertainty. In the middle rows, moderate-density scenarios display an increased number of agents. Here, interactions among vehicles become more frequent, creating added complexity for trajectory prediction. In the bottom row, high-density scenes pose significant challenges for trajectory prediction models. These crowded urban intersections and multi-agent interactions are underrepresented in standard datasets. The large number of dynamic agents in these scenarios compounds uncertainty, making it harder for the model to produce accurate predictions. Yet, these are precisely the scenarios that are most critical for ensuring safe autonomous driving. Our SSTP method addresses this imbalance head-on by emphasizing high-density samples. Through a more balanced yet compact selection of training data, the model becomes well-prepared for both common and complex scenarios. As a result, SSTP bolsters model robustness in high-density environments, leading to more reliable trajectory predictions in real-world urban traffic conditions.

## LLM USAGE

We used ChatGPT solely for grammar correction and LaTeX formatting. They were not involved in research ideation, experiment design, or data analysis. All scientific contributions, methodology, and results are entirely the work of the authors.

