# OpenReview forum: "SSTP: Efficient Sample Selection for Trajectory Prediction"
_ICLR.cc/2026/Conference — ICLR 2026 Conference Withdrawn Submission_

### Official Review · Reviewer_ingr · 2025-10-30

**Soundness:** 2
**Presentation:** 3
**Contribution:** 2
**Rating:** 4
**Confidence:** 5

**Summary:**

This paper introduces SSTP, a sample selection framework for trajectory prediction. The primary motivation is to address two challenges in existing large-scale datasets: the high computational cost of training and the imbalance where common, low-density scenarios dominate over rare, safety-critical high-density ones. The proposed method consists of two stages. First, it partitions the dataset based on scene density (number of agents) and pre-trains a model to extract gradient information. Second, it uses a submodular selection objective with these gradient-based scores to select a compact and representative subset, while explicitly up-sampling high-density scenarios. Experiments on the Argoverse 1 and 2 datasets show that training on a 50% subset selected by SSTP can achieve comparable performance to training on the full dataset, while significantly improving performance in high-density scenes.

**Strengths:**

* The paper tackles a relevant and practical problem in trajectory prediction: the long-tail distribution of driving scenarios. Improving performance on rare but safety-critical, high-density scenes is a valuable contribution.

* The proposed framework, SSTP, is novel for this domain. Applying a principled, gradient-based submodular selection approach to create a compact and balanced dataset for trajectory forecasting is a new and interesting direction.

* The experimental results are promising, showing consistent improvements in high-density scenarios across multiple datasets and models (HiVT, HPNet). This demonstrates the effectiveness of the proposed density-balancing strategy.

* The work shows strong generalizability. The subset selected by SSTP using one model backbone transfers well when used to train other, more powerful models, saving computation time on the more expensive model.

**Weaknesses:**

* Motivation for sample selection is not entirely convincing. The paper highlights two goals: reducing training time and improving long-tail performance.

    * Regarding training time, a 2x speedup by using 50% of the data is a relatively modest gain, especially when considering the non-negligible computational overhead of the selection process itself. Training times for trajectory prediction models (e.g., ~8 hours for HiVT , ~300 hours for HPNet ) are not as prohibitive as in other domains like large language models, or diffusion models for image and video generation, which makes the efficiency argument less compelling. Also, sample selection methods typically aim for much larger data reduction (e.g., 90-99%) to justify the method and implementation complexity by a larger gain in training time decreasing. However the results for this higher filtering rates in table 9 shows the performance drop is very high and reduces the contribution of this work.

    * If the main goal is improving long-tail performance, the paper lacks comparisons to the significant body of work that directly addresses this issue in trajectory prediction (e.g., using contrastive learning or other re-balancing techniques ), some of which are cited in the related work section.

* The definition of "agent density" is overly simplistic. The paper uses the total number of agents in a scene as a proxy for interaction density. This heuristic could be misleading, as high agent counts might result from parked cars or agents that are far away and not interacting with the vehicle of interest. The fact that the "low-density" bucket includes scenes with up to 40 agents seems to support this, as it's unlikely a driver would interact with so many agents in a short 5-second interval. A more refined metric, such as counting only moving agents within a certain proximity, could provide a more accurate measure of scene complexity.

* Performance degradation on common scenarios. The results show that while SSTP improves performance on high-density cases, it leads to a noticeable drop in performance on more common low-density scenarios. For example, in Table 8, the HPNet model's minADE on low-density scenes (Agent<40) increases from 0.611 to 0.630 when trained on the 50% SSTP subset. For many practical applications, this trade-off might not be acceptable, and one might prefer to train for twice as long to achieve better performance on the majority of cases.

* Potential redundancy of sample selection. Publicly available datasets like Argoverse are typically already curated to contain a variety of interesting and non-trivial driving scenarios. This pre-selection might limit the potential gains of an additional sample selection layer, making it harder to find and discard truly redundant samples.

**Questions:**

1. Could you elaborate on the choice of using the raw number of agents as the metric for scene density? Have you considered alternative, potentially more robust heuristics, such as filtering out non-moving agents or agents outside a certain proximity, to get a more accurate measure of interaction complexity?

2. The results indicate a performance drop in low-density scenarios in exchange for gains in high-density ones. Could you comment on this trade-off? Is this an inherent limitation of the approach, or could the framework be adjusted to mitigate this degradation while preserving the benefits for rare scenarios?

3. The paper's contribution to improving long-tail performance is clear. However, why were alternative methods designed specifically for long-tail trajectory prediction not included as baselines for comparison, particularly regarding performance in high-density scenarios?

4. Given that the training time is only halved, and the selection process itself adds computational overhead, could you further justify the motivation from a training efficiency perspective, especially when sample selection literature often targets much larger (10x-100x) speedups?

5. I would be very interested to see results of applying this methodology to fields other than trajectory prediction, where training models are indeed very expensive and data has a lower quality, allowing for higher gains in selecting interesting samples.

---

### Official Review · Reviewer_b5Zf · 2025-11-01

**Soundness:** 3
**Presentation:** 3
**Contribution:** 2
**Rating:** 4
**Confidence:** 4

**Summary:**

This paper proposes SSTP, a two-stage sample selection framework that constructs a compact yet density-balanced dataset for trajectory prediction. It consists of two stages: (i) first partition the data by scene density; and (ii) select a compact and density-balanced subset via gradient-based scores and a submodular objective. The goal is to reduce training time and mitigate long-tail imbalance. On Argoverse 1 and 2 Datasets and several backbones (HiVT, HPNet, QCNet, DeMo), SSTP claims comparable average metrics to full-data training, and improving high-density performance with around 50% budget.

**Strengths:**

1. SSTP achieves consistent empirical gains at 40-60% budget, and beats augmenting, weighting, epoch-wise reselection, random, and high-density+random in terms of minADE, minFDE and MR.
2. The paper has comprehensive ablation studies, such as Table 5-12.

**Weaknesses:**

1. The paper uses a simple metric of agent count as density to divide the data into bins, but uses a relatively complicated construction for gradient features to select data within bins. One might wonder (1) if there are other possible definitions of density, and (2) if there are other possible constructions of data selection criterion within bins.
2. The density imbalance in autonomous driving dataset is a well known problem and it is known that re-balancing the data during training might hurt the overall performance on the entire dataset, as shown in Table 3 and 4.
3. In practice, for example in an autonomous driving company, practitioners would mostly likely first train on the full dataset and only then apply SSTP to potentially improve the model. In this sense, SSTP is usable primarily a posteriori. It offers a possible quality boost rather than a budget saver.

**Questions:**

Have you tried other possible feature formulations to select data within bins?

---

### Official Review · Reviewer_WSH7 · 2025-11-01

**Soundness:** 3
**Presentation:** 3
**Contribution:** 2
**Rating:** 4
**Confidence:** 5

**Summary:**

The paper aims to address an important problem of reducing dependency on large-scale datasets in trajectory prediction, particularly under imbalanced data distributions.

**Strengths:**

The proposed SSTP framework aims to construct compact and balanced training subsets via density-based partitioning and gradient-informed sample selection.

**Weaknesses:**

While the motivation is relevant and the experimental results show improved efficiency, I have several conceptual and technical concerns that prevent me from supporting acceptance in its current form.

My main concerns are as follows:
The method relies heavily on partitioning data based solely on the number of agents per scene. This definition of "scene density" appears oversimplified, as it does not account for other critical factors such as interaction complexity, road topology, agent behaviors, or environmental conditions. In many real-world scenarios, a high agent count (e.g., in a parking lot) may not correspond to high risk, while sparse scenes can still be highly complex. A more nuanced and context-aware measure of scene difficulty is needed to justify the selection strategy.

The notion of "long-tail" in trajectory prediction is not rigorously defined or statistically established. The authors should provide a clearer formulation of the long-tail phenomenon specific to autonomous driving, supported by dataset statistics (e.g., distribution plots, tail indices) rather than qualitative descriptions. This would help align the problem statement with the proposed solution.

Although the paper motivates the work with the long-tail challenge, it does not sufficiently demonstrate that SSTP effectively addresses tail scenarios. Evaluation on dedicated corner-case benchmarks or tail-oriented splits is missing. A comparison with existing long-tail learning methods adapted to trajectory prediction would also strengthen the validity of the approach.

The technical components of SSTP, like density partitioning and gradient-based submodular selection, appear to be combinations of existing ideas rather than fundamentally novel contributions. The authors should more clearly differentiate their method from related work in coreset selection or gradient-based sample pruning. Additionally, there is no sensitivity analysis on key hyperparameters, such as the similarity threshold in the submodular function, which raises concerns about robustness and generalizability.

The requirement of a pretrained model for gradient extraction introduces non-negligible computational overhead, partly counteracting the efficiency gains of data selection. Moreover, submodular optimization in high-dimensional gradient spaces may not scale efficiently. These practical limitations should be acknowledged and quantitatively analyzed.

The experimental evaluation is limited to the Argoverse 1 and 2 datasets, which share similar characteristics. To establish generalizability, validation on more diverse benchmarks such as nuScenes or Waymo Open Motion Dataset is essential. Additionally, comparisons are made only against simple baselines (random, clustering, herding). Including recent advanced selection methods like GradMatch or CRAIG would offer a more convincing and competitive assessment.

The writing structure could be improved by moving the related work section earlier to better contextualize the contribution.

Furthermore, the analysis would benefit from including failure cases or scenarios where SSTP does not perform well, offering deeper insight into its limitations.

In summary, while the idea of data-efficient training for trajectory prediction is valuable and timely, the current manuscript lacks sufficient methodological novelty, rigorous evaluation, and generalizability to meet the bar for ICLR. I encourage the authors to address these points in a future revision, particularly by refining the notion of scene complexity, expanding experiments across datasets and baselines, and providing more thorough analytical validation.

**Questions:**

See the last section.

---

### Official Review · Reviewer_MR4w · 2025-11-03

**Soundness:** 2
**Presentation:** 2
**Contribution:** 2
**Rating:** 2
**Confidence:** 3

**Summary:**

This paper proposes SSTP, a framework designed to improve data efficiency and scene-density balance in trajectory prediction. The authors observe that existing large-scale trajectory prediction datasets are heavily imbalanced, with low-density scenarios dominating and high-density cases underrepresented. SSTP tackles this issue through two-stage process: density-based partitioning of the dataset and gradient-based submodular selection to identify representative samples within each partition. Experiments on Argoverse 1 and Argoverse 2 show that SSTP achieves comparable performance to full-dataset training while reducing training cost and improving performance in high-density scenarios.

**Strengths:**

1. The paper clears out a key limitation in current imbalanced trajectory prediction dataset.
2. The method is systemically validated across multiple models and two benchmark datasets.
3. The paper is clearly written and easy to follow.

**Weaknesses:**

1. The paper raises an important problem regarding scene-density imbalance in trajectory prediction, but the proposed approach offers only a partial solution. SSTP mainly combines known concepts such as density-based partitioning and submodular selection rather than introducing a fundamentally new modeling idea. As a result, the contribution lacks novelty.

2. The comparison in Table 3 seems unfair and potentially misleading. Since high-density scenarios are few and mostly retained while low-density ones are pruned, performance improvement in dense scenes is somewhat expected. However, the noticeable drop in low-density performance raises doubts about the true benefit of SSTP, as improving performance in complex cases while degrading simpler ones limits its overall practical value.

3. The definition of scene density based solely on the number of agents may oversimplify scene complexity. A high agent count does not always indicate complex interactions, and low-density scenes can still involve diverse or challenging trajectories. Considering additional measures such as interaction diversity, trajectory variance, or spatial proximity could lead to a more accurate and interpretable characterization of scenario complexity, thereby improving the effectiveness of the density-aware selection process.

4. The paper lacks qualitative examples or visualizations of the prediction results. Although the authors claim that SSTP improves performance in scenarios with many agents, it is unclear how the predicted trajectories differ in these complex, high-density scenes. Visual comparisons or representative examples would help illustrate the practical impact of the proposed selection strategy.

5. Evaluation on a more diverse dataset such as Waymo Open Motion Dataset would strengthen the generality of the method.

**Questions:**

In Table 5, the ablation study compares the effects of using partitioning submodular selection, and the full SSTP framework. However, the paper does not clearly explain how each component contributes to performance across different scene densities. Could the authors provide model performance results broken down by the number of agents for each ablation variant and interpret how partition and submodular selection influence each density scenarios?

---

### Note · Authors · 2025-11-12

I have read and agree with the venue's withdrawal policy on behalf of myself and my co-authors.